# Class-Dependent Label-Noise Learning with Cycle-Consistency Regularization

**De Cheng**[1],[*] **Yixiong Ning**[1],[*] **Nannan Wang**[1],[†]
**Xinbo Gao**[2]**, Heng Yang**[3]**, Yuxuan Du**[4]**, Bo Han**[5]**, Tongliang Liu**[6]
[1] Xidian University, [2] Chongqing University of Posts and Telecommunications,
[3] Shenzhen AiMall Tech. Co., Ltd., [4] JD Explore Academy,
[5] Hong Kong Baptist University, [6] TML Lab, The University of Sydney.

## Abstract

In label-noise learning, estimating the transition matrix plays an important role in building statistically consistent classifier. Current state-of-the-art consistent estimator for the transition matrix has been developed under the newly proposed *sufficiently scattered* assumption, through incorporating the minimum *volume* constraint of the transition matrix $T$ into label-noise learning. To compute the *volume* of $T$, it heavily relies on the estimated *noisy class posterior*. However, the estimation error of the *noisy class posterior* could usually be large as deep learning methods tend to easily overfit the noisy labels. Then, directly minimizing the *volume* of such obtained $T$ could lead the transition matrix to be poorly estimated. How to reduce the side-effects of the inaccurate *noisy class posterior* remains unsolved. In this paper, we creatively propose to estimate the transition matrix under a forward-backward cycle-consistency regularization, of which we have greatly reduced the dependency of estimating the transition matrix $T$ on the *noisy class posterior*. Extensive experimental results consistently justify the effectiveness of the proposed method, on reducing the estimation error of the transition matrix and greatly boosting the classification performance.

## 1  Introduction

Deep learning based algorithms rely heavily on large-scale annotated training data. However, it is often extremely costly and sometimes even infeasible to accurately annotate such massive dataset [13]. An usual way to address this issue is to collect large-scale training data using cheap methods, e.g., from the crowd-sourcing platform [36] or online query search engine [3], which could inevitably yield label noise. Training deep model with label noise can significantly degrade the performance [13, 1, 28]. Therefore, how to mitigate the side-effects of the label noise automatically becomes very important, and it has drawn increasing attention recently.

Recent studies show that estimating the transition matrix plays an important role in building statistically consistent classifiers for label-noise learning, since the transition matrix can well model the noise generation process. These methods guarantee that classifier learned from the noisy data could approach to the optimal classifier defined on the clean risk asymptotically as the size of the noisy training data increases [13]. The basic principle is that the clean class posterior $P(\mathbf{Y}|X = \mathbf{x})$[1] can be inferred by the noisy class posterior $P(\bar{\mathbf{Y}}|X = \mathbf{x})$ and the transition matrix $T(\mathbf{x})$, where

---

[*]Equation contribution.
[†]Corresponding author.
[1]We define $P(\mathbf{Y}|X = \mathbf{x}) = [P(Y = 1|X = \mathbf{x}), \ldots, P(Y = C|X = \mathbf{x})]^\top$ where $C$ represents the number of classes.

36th Conference on Neural Information Processing Systems (NeurIPS 2022).

$T_{ij}(\mathbf{x}) = P(\bar{Y} = j | Y = i, X = \mathbf{x})$, i.e., $P(\bar{\mathbf{Y}} | X = \mathbf{x}) = T(\mathbf{x})P(\mathbf{Y}|X = \mathbf{x})$. This paper focuses on the common class-dependent and instance-independent label noise, i.e., $T(\mathbf{x}) = T$. Usually, the transition matrix $T$ is unidentifiable and hard to learn without additional assumptions [37, 5, 33].

In the literature, many methods try to estimate the transition matrix under the *anchor point* assumption, which assumes that there exist some instances belonging to a specific class almost surely [33, 16, 22]. However, the *anchor point* assumption can not be always satisfied [24, 39]. Therefore, methods aiming to develop statistically consistent classifiers without anchor points have been proposed [33, 13]. Among them, the state-of-the-art method (VolMinNet [13]) proposed by far the mildest *sufficiently scattered* assumption, where the *anchor point* assumption is its special case [13]. They proved that the optimal $T$ is the one with minimum *volume* enclosing the noisy class posterior of training examples. Hence, they proposed to incorporate the minimum *volume* constraint of $T$ into label-noise learning. To compute the *volume* of $T$, the existing method heavily relies on the estimated *noisy class posterior*. However, the estimation error of the *noisy class posterior* could usually be large as the deep learning methods tend to easily overfit the noisy labels, especially under the settings of limited training samples [37]. This could lead the transition matrix to be poorly estimated when minimizing the *volume* calculated by using the learned noisy class posteriors. Therefore, how to reduce the side-effects of the inaccurate *noisy class posterior* has become a major concern for making use of the sufficiently scattered assumption.

To address this issue, we propose a forward-backward cycle-consistency regularized algorithm to estimate the transition matrix $T$, of which we could greatly reduce the dependency of estimating $T$ on using the *noisy class posterior* to calculate its volume, and further build a statistically consistent classifier. Specifically, we show that minimizing the *volume* of the transition matrix $T$ is equal to maximizing the *volume* of corresponding clean class posterior. More importantly, maximizing the *volume* of the clean class posterior could avoid the side-effects of the inaccurate noisy class posterior probability on estimating $T$.

To be more specific, we will further exploit the equation $P(\bar{\mathbf{Y}} | X = \mathbf{x}) = TP(\mathbf{Y}|X = \mathbf{x})$, where $P(\bar{\mathbf{Y}} | X = \mathbf{x}) \in \mathbb{R}^C$, $T \in \mathbb{R}^{C \times C}$, $P(\mathbf{Y}|X = \mathbf{x}) \in \mathbb{R}^C$, and $C$ is the number of classes. Note that $P(\bar{\mathbf{Y}} | X = \mathbf{x})$ can be estimated by exploiting the noisy data. For simplicity, we re-write the above equation as $A = TB$ for the following derivation, where $A$ and $B$ represent $P(\bar{\mathbf{Y}} | X = \mathbf{x})$ and $P(\mathbf{Y}|X = \mathbf{x})$, respectively. We adopt the commonly used determinant operation to measure the *volume* of $T$, i.e., $\det(T)$. As only the square matrix has determinant, we have $\det(AA^\top) = \det(TBB^\top T^\top) = \det(TT^\top)\det(BB^\top)$. As $P(\bar{\mathbf{Y}} | X = \mathbf{x})$, i.e., $A$, can be estimated by the given noisy data, the determinant $\det(AA^\top)$ is constant. Then, minimizing $\det(T)$ (or $\det(TT^\top)$) equals maximizing $\det(BB^\top)$. Therefore, we can replace the minimum *volume* constraint of $T$ by the maximum *volume* constraint of the clean class posterior $P(\mathbf{Y}|X = \mathbf{x})$.

Maximizing the *volume* of the clean class posterior $P(\mathbf{Y}|X = \mathbf{x})$ can be achieved without directly relying on the estimated noisy class posterior probability. Specifically, we project the $C$-dimensional simplex, which has the largest *volume*, from the noisy class posterior space onto the clean class posterior space by a *diagonally dominant column stochastic* matrix $T'$ [13]. Let denote the projected clean class posteriors as the backward clean class posteriors $P'(\mathbf{Y}|X = \mathbf{x})$. They will make up the largest *volume* in the sense that they correspond to the $C$-dimensional simplex in the noisy class posterior space. Afterwards, we guide the clean class posterior $P(\mathbf{Y}|X = \mathbf{x})$ learned by the neural network to match to $P'(\mathbf{Y}|X = \mathbf{x})$. Therefore, the *volume* of the learned clean class posteriors could be maximized to encourage the transition matrix $T$ to converge to its optimal solution. Besides, we further build the cycle-consistency regularization between the forward and backward transition matrices $T$ and $T'$ to better learn the clean class posterior probability.

Our main contributions are summarized as follows: 1) We propose to estimate the transition matrix $T$ under a forward-backward cycle-consistency regularization, of which we could greatly reduce the dependency of minimizing the volume of the transition matrix $T$ on the estimated *noisy class posterior*; 2) We show that such cycle-consistency regularization could help to minimize the *volume* of the transition matrix $T$ without directly exploiting the estimated noisy class posterior, which encourages the estimated transition matrix $T$ to converge to the optimal solution; 3) Experimental results on four datasets (two synthetic and two real-world datasets) with different label-noise settings consistently justify the effectiveness of the proposed method, on reducing the estimation error of the transition matrix and greatly improving the classification performance.

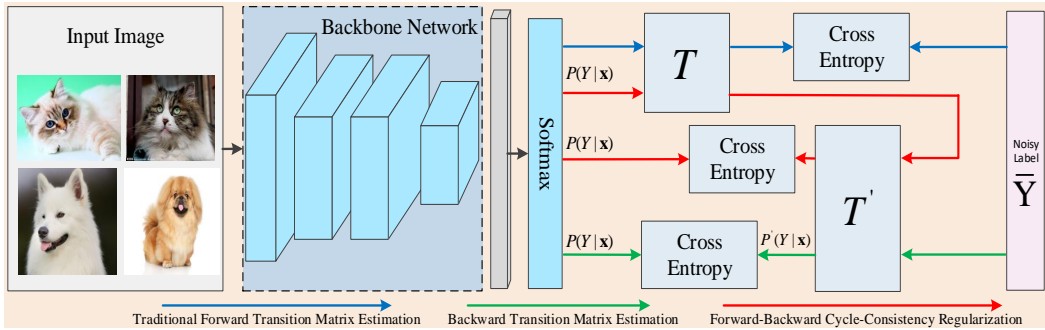

Figure 1: Overview of our method. We optimize the proposed method in an end-to-end manner with three objectives simultaneously. The blue line represents the traditional forward transition matrix $T$ estimation, the green line means the backward transition matrix $T'$ estimation. The red line means the cycle-consistency regularization between the forward and backward transition matrices.

## 2 Estimating Transition Matrix for Label-Noise Learning

**Problem Setting.** Let $D$ be the distribution of a pair of random variables $(\mathbf{X}, \mathbf{Y}) \in (\mathcal{X} \times \mathcal{Y})$, where $\mathbf{X}$ and $\mathbf{Y}$ denote the variable of instances and the corresponding labels, $\mathcal{X} \in \mathbb{R}^{d \times N}$ represents the instance feature space, $d$ is the feature dimension, $N$ is the total number of instances, $\mathcal{Y} \in \{1, \cdots, C\}$ denotes the label space, and $C$ is the size of the classes. However, in some real-world scenarios, it is often expensive or even infeasible to accurately draw large-scale training data independently from the clean distribution $D$.

Let $\bar{D}$ be the distribution of a pair of random variables $(\mathbf{X}, \bar{\mathbf{Y}}) \in (\mathcal{X} \times \mathcal{Y})$, where $\bar{\mathbf{Y}}$ denotes the variable of noisy labels. In learning with noisy labels, we denote the training dataset as $\bar{\mathcal{D}} = \{(\mathbf{x}_i, \bar{y}_i)\}_{i=1}^{N}$ which is independently drawn from the noisy distribution $\bar{D}$. The goal is to learn a robust classifier by exploiting the noisy dataset $\bar{\mathcal{D}}$, which approximates well to the optimal classifier defined on the clean data.

**Transition Matrix** plays an important role in building statistically consistent classifier through the "loss correction" strategy. It explicitly models the label-noise generation process from clean data distribution to the noisy data distribution, i.e., $T_{ij}(\mathbf{x}) = P(\bar{Y} = j | Y = i, X = \mathbf{x})$, and $T(\mathbf{x}) \in \mathbb{R}^{C \times C}$. It represents the probability that an instance $\mathbf{x}$ having the clean label $Y = i$ is mislabeled as the noisy label $\bar{Y} = j$. Intuitively, the clean class posterior probability $P(\mathbf{Y}|\mathbf{x})$ can be inferred by the transition matrix $T_{ij}(\mathbf{x})$ and the noisy class posterior probability $P(\bar{\mathbf{Y}}|\mathbf{x})$, i.e., $P(\bar{\mathbf{Y}}|\mathbf{x}) = T(\mathbf{x})P(\mathbf{Y}|\mathbf{x})$. Note that, in the above equation, the noisy class posterior $P(\bar{\mathbf{Y}}|\mathbf{x})$ can be estimated by the noisy data. The transition matrix $T(\mathbf{x})$ is generally unidentifiable and very hard to estimate without additional assumptions. In this paper, we focus on the widely studied class-dependent and instance-independent label noise, i.e, $T(\mathbf{x}) = T, T \in \mathbb{R}^{C \times C}$.

Current state-of-the-art method (VolMinNet [13]) proposes to estimate the class-dependent transition matrix $T$ under the *sufficiently scattered* assumption, where the traditional *anchor point* assumption is its special case. They proved that the optimal $T$ is the one with minimum *volume* enclosing the noisy class posterior of the training examples. However, in this work, we resort to address its equivalence problem, i.e., maximizing the *volume* of corresponding clean class posterior probability, which could reduce the side-effect of the estimation error of the noisy class posterior on estimating $T$. Therefore, to maximize the *volume* of $P(\mathbf{Y}|X)$, we propose the forward-backward cycle-consistency regularization, and further to consistently estimate $T$. Specifically, as shown in Fig 1, our proposed method simultaneously optimize three objectives: the cross entropy loss between the given noisy label $\bar{y}$ and the estimated noisy class posterior; the cross entropy loss between the backward estimated clean class posteriors and the predicted clean class posteriors by the neural network; the cycle-consistency regularization to minimize the approximation error between $P(\mathbf{Y}|X)$ and $T'TP(\mathbf{Y}|X)$. The backward and cycle-consistency regularization help to minimize the volume of $T$ indirectly without exploiting the noisy class posterior, and further encourage the estimated $T$ to converge to its optimal solution.

## 2.1 Forward Transition Matrix Estimation

Given the input instance $\mathbf{x}$, the probability of observing a noisy label $\bar{Y}$ can be inferred as,

$$P(\bar{Y} = j|\mathbf{x}) = \sum_{i=1}^{C} P(\bar{Y} = j|Y = i, \mathbf{x})P(Y = i|\mathbf{x}) = \sum_{i=1}^{C} P(\bar{Y} = j|Y = i)P(Y = i|\mathbf{x})$$
$$= \sum_{i=1}^{C} T_{ij}P(Y = i|\mathbf{x}). \tag{1}$$

We can rearrange Eq. 1 as $P(\bar{\mathbf{Y}}|\mathbf{x}) = TP(\mathbf{Y}|\mathbf{x})$ under the class dependent and instance independent assumption, where $P(\bar{\mathbf{Y}}|\mathbf{x}) = [P(\bar{Y} = 1|\mathbf{x}), ..., P(\bar{Y} = C|\mathbf{x})]^{\top}$ and $P(\mathbf{Y}|\mathbf{x}) = [P(Y = 1|\mathbf{x}), ..., P(Y = C|\mathbf{x})]^{\top}$ are the noisy class posterior probability and the clean class posterior probability, respectively. To learn the clean class posterior $P(\mathbf{Y}|X)$, we design a neural network $f(\mathbf{x}; \mathbf{w})$ parameterized with $\mathbf{w}$, which projects the input instance $\mathbf{x}$ onto $C$ classes with a probability output. The transition matrix $T$ should be trainable *diagonally dominant column stochastic* matrix, i.e., $T \in [0, 1]^{C \times C}, \sum_{i=1}^{C} T_{ij} = 1$ and $T_{ii} > T_{ji}$ for any $i \neq j$. Naturally, we could use $Tf(\mathbf{x}; \mathbf{w})$ to learn the noisy class posterior probability $P(\bar{\mathbf{Y}}|X)$. VolMinNet [13] shows us that, if $Tf(\mathbf{x}; \mathbf{w})$ models $P(\bar{\mathbf{Y}}|X)$ perfectly and the transition matrix $T$ has the minimum volume, $T$ will converge to the optimal transition matrix and the classifier $f(\mathbf{x}; \mathbf{w})$ will converge to the optimal $P(\mathbf{Y}|X)$. In this paper, we introduce the cycle-consistency regularization to indirectly reduce the volume of $T$, without exploiting the estimated noisy class-posterior probability which usually has a large estimation error [37].

Given the training dataset $\bar{\mathcal{D}} = \{(\mathbf{x}_i, \bar{y}_i)\}_{i=1}^{N}$, we optimize the empirical risk by jointly optimizing the transition matrix $T$ and the consistent classifier $f(\mathbf{x}; \mathbf{w})$ for label-noise learning. Specifically, we minimize the approximation error between the inferred noisy class-posterior probability $Tf(\mathbf{x}; \mathbf{w})$ and the given noisy label $\bar{y}$ as follows,

$$\min_{\mathbf{w}, T} \mathcal{L}_1(\mathbf{w}, T) = -\frac{1}{N} \sum_{i=1}^{N} \bar{y}_i \log(T \cdot f(\mathbf{x}_i; \mathbf{w})), \tag{2}$$

where $N$ is the number of training examples. Note that, in Eq. 2, both the transition matrix $T$ and the output of classifier $f(\mathbf{x}_i; \mathbf{w})$ are not given, which makes the transition matrix $T$ unidentifiable without any assumptions. To address this issue, VolMinNet [13] has been proposed by minimizing the volume of the transition matrix. It has been proved that VolMinNet theoretically guarantees the consistencies of both the transition matrix and the classifier.

## 2.2 Cycle-Consistency Regularization

When minimizing the volume of the transition matrix, VolMinNet exploits the estimated noisy class posteriors to calculate the volume. Yao et al. [37] has discussed that the estimation error of the noisy class posterior is usually large due to the randomness of label noise. This would lead the transition matrix to be poorly estimated by VolMinNet when the noisy training sample is limited. We will address this issue by proposing a cycle-consistency regularization method instead of minimizing the volume of the transition matrix directly.

As discussed in the Introduction, minimizing the volume of the transition matrix is equivalent to maximizing the volume of the clean class posterior. We show that we could directly use the noisy labels to maximize the volume of the clean class posterior rather than using the estimated noisy class posterior to calculate the volume of the transition matrix and then minimizing it. Intuitively, we maximize the volume of the learned clean class posteriors by matching it with the projected $C$-dimensional simplex in the noisy class posterior which also needs to estimate.

More specifically, we first transform the one-hot noisy label into the input noisy class posterior probability by using the SoftMax function, denoting it as $\text{SoftMax}(\bar{y}_i)$. Then, we use $T'$ (an approximation of the inverse of the transition matrix) to project it onto the clean class posterior space. Since the true clean class posterior is unobservable, here we use the output of the designed

neural network $f(\mathbf{x}_i; \mathbf{w})$ as its approximation. Finally, we minimize the following cross-entropy loss between $T'\text{SoftMax}(\bar{y}_i)$ and $f(\mathbf{x}_i; \mathbf{w})$, to optimize the parameters of $\mathbf{w}$ and $T'$ simultaneously,

$$\min_{\mathbf{w}, T'} \mathcal{L}_2(\mathbf{w}, T') = -\frac{1}{N} \sum_{i=1}^{N} f(\mathbf{x}_i; \mathbf{w}) \log(T' \cdot \text{SoftMax}(\bar{y}_i)). \tag{3}$$

Note that this objective will help maximize the volume of the learned clean class posterior as $T'\text{SoftMax}(\bar{y}_i)$ has the largest volume in the sense that the noisy labels will make up the $C$-dimensional simplex. Note also that $T'$ should be initialized by and maintained as a *diagonally dominant column stochastic* matrix, i.e., $T' \in [0, 1]^{C \times C}, \sum_{i=1}^{C} T'_{ij} = 1$ and $T'_{ii} > T'_{ji}$ for any $i \neq j$.

At present, we can obtain the transition matrices from both the forward and backward ways, i.e., $T$ and $T'$, by minimizing $\mathcal{L}_1$ and $\mathcal{L}_2$, respectively. Intuitively, we should also build an "indirect" cycle-consistency through minimizing the difference between $P(\mathbf{Y}|X = \mathbf{x})$ and $T'(TP(\mathbf{Y}|X = \mathbf{x}))$, where "indirect" means that we would make use of the invertible relationship between these two matrices $T$ and $T'$ indirectly through the original and circularly computed clean class-posterior. Specifically, the cycle-consistency regularization can be expressed as,

$$\min_{\mathbf{w}} \mathcal{L}_3(\mathbf{w}; T, T') = -\frac{1}{N} \sum_{i=1}^{N} f(\mathbf{x}_i; \mathbf{w}) \log(T'(T \cdot f(\mathbf{x}_i; \mathbf{w}))). \tag{4}$$

Note that, in Eq. 4, we just optimize the parameter $\mathbf{w}$ in the network classifier $f(\mathbf{x}_i; \mathbf{w})$, while keep $T$ and $T'$ be constant. This is because that our ultimate goal is to learn good clean class posteriors. If the transition matrices are well learned, the estimated clean class posterior should satisfy this constraint.

Based on the above descriptions, the overall objective function can be expressed as,

$$\mathcal{L} = \mathcal{L}_1(\mathbf{w}, T) + \mathcal{L}_2(\mathbf{w}, T') + \lambda \mathcal{L}_3(\mathbf{w}; T, T'), \tag{5}$$

where $\lambda$ is the trade-off hyper-parameter to balance the two transition matrix learning objectives with the cycle-consistency regularization term. We optimize all the model parameters simultaneously in an end-to-end manner.

## 2.3 Theoretical Analysis

In this section, we justify that the proposed method helps to minimize the *volume* of the transition matrix $T$ without directly relying on the learned noisy class posterior, which does not prevent the estimated $T$ from converging to its optimal solution.

As theoretically proved in VolMinNet [13], the optimal transition matrix $T$ is the one with minimum *volume* enclosing the noisy class posterior of the training examples under *sufficiently scattered* assumption. VolMinNet thereby theoretically guarantees the consistencies of both the transition matrix and the classifier by minimizing the volume of the transition matrix. Under the same *sufficiently scattered* assumption, our proposed method also theoretically guarantees the consistencies of both the transition matrix and the classifier because, in Introduction, we show that minimizing the *volume* of the transition matrix is equal to maximizing the *volume* of the clean class posterior. Importantly, our method could help reduce the side-effects of the estimation error of the noisy class-posterior probability.

Specifically, our proposed framework as shown in Eq. 5, simultaneously optimize the forward and backward transition matrices $T$ and $T'$. Especially during $T'$ estimation, the input is the given one-hot noisy label which corresponds to the simplex having the maximum *volume*. When they pass through the diagonally dominant matrix $T'$, the projected clean class-posterior probability should intuitively has a large *volume*. Then, minimizing the cross entropy between the learned clean class posteriors and the projected clean class posteriors could help to maximize the *volume* of the learned clean class posteriors. Note that the cycle-consistency constraint on $T$ and $T'$ as illustrated in Eq. 4 further helps to learn better clean class posterior as the projected clean class posterior is more accurate.

## 3 Experiments

In this section, we introduce the experiment setup, including datasets, noise types, and implementation details. We compare our proposed method with the state-of-the-art algorithms on two synthetic and

Table 1: Comparison with state-of-the-art methods on CIFAR-10 and CIFAR-100 datasets. The mean and standard deviation computed over five runs are presented. "Sym-xx%" means the noise rate is xx% and noise type is "Symmetry".

| Method | Cifar-10 | | | CIFAR-100 | | |
|---|---|---|---|---|---|---|
| | Sym-20% | Sym-40% | Sym-60% | Sym-20% | Sym-40% | Sym-60% |
| CE (baseline) | 84.58± 0.18 | 80.78± 0.38 | 68.31± 0.33 | 51.93 ±0.39 | 40.11± 0.86 | 25.81± 0.74 |
| GCE [40] | 89.31 ±0.07 | 86.61± 0.23 | 79.40± 0.41 | 66.07± 0.24 | 59.03± 0.21 | 45.68± 0.39 |
| PeerLoss [17] | 88.78 ±0.18 | 84.87± 0.15 | 75.28± 0.31 | 57.34± 0.34 | 43.39± 0.33 | 28.66± 0.67 |
| Co-teaching [6] | 85.76 ±0.26 | 83.12± 0.31 | 70.89± 1.06 | 56.83± 0.28 | 43.38± 0.51 | 28.04± 0.69 |
| Co-teaching++ [38] | 86.39±0.33 | 83.80±0.30 | 72.51±0.46 | 57.64±0.34 | 44.28±0.83 | 29.60±1.16 |
| T-Revision [33] | 88.01±0.16 | 84.52±0.11 | 71.53±0.82 | 62.66±0.53 | 55.25±0.36 | 39.94±1.28 |
| VolMinNet [13] | 89.69±0.19 | 85.46±0.19 | 73.55±0.78 | 64.70±0.60 | 56.25±0.45 | 41.06±0.45 |
| DualT [37] | 89.88±0.13 | 86.23±0.64 | 72.21±1.67 | 65.75±0.38 | 56.80±0.18 | 42.56±0.55 |
| T-For ($T$) | 89.53±0.11 | 85.38±0.13 | 73.01±0.54 | 64.23±0.64 | 56.02±0.39 | 40.89±0.37 |
| T-Back ($T'$) | 88.40±0.12 | 84.97±0.16 | 73.12±0.79 | 63.39±0.62 | 54.96±0.43 | 41.15±0.82 |
| $T + T'$ | 89.64±0.16 | 85.47±0.32 | 73.39±0.40 | 64.95±0.91 | 56.36±0.51 | 41.94±0.43 |
| Ours | **90.44**±0.19 | **87.30**±0.25 | **81.01**±0.25 | **67.74**±0.17 | **61.71**±0.20 | **49.30**±0.82 |

two real-world noisy datasets, followed by an ablation study to analyze the experimental results and some useful hyper-parameters.

## 3.1 Experiment Setup

**Datasets**. Extensive experiments are conducted on two manually corrupted datasets with different noisy types (i.e., CIFAR-10 [9], CIFAR-100 [9]) and two real-world noisy datasets ( i.e., Clothing1M [34] and Food-101N [10]), to demonstrate the effectiveness of the proposed method. Both CIFAR-10 and CIFAR-100 contain 60K images of size $32 \times 32$, of which 50K images constitute the training set and 10K images for testing set, while CIFAR-10 contains 10 classes, and CIFAR-100 contains 100 classes. Clothing1M is a real-world noisy dataset that contains 1M images with about 38.46% noisy labels for training and 10K images with clean labels for testing. Food-101N is also a real-world noisy dataset which contains 310K images with about 19.66% noisy labels for training and 55K images with clean labels for testing.

**Noisy type**. For CIFAR-10 and CIFAR-100, we manually corrupted the training set according to the ground-truth transition matrices. Specifically, we conducted experiments using three commonly used noisy types: (1) Symmetry flipping [20]; (2) Asymmetry flipping [12]; (3) Pair flipping [8];

## 3.2 Implementation Details

For fair comparisons, all our experiments are performed on NVIDIA GeForce RTX 3090, and implemented on the same PyTorch platform. For CIFAR-10 and CIFAR-100, the backbone we used is ResNet-34. We train the classification network $f(\mathbf{x}_i; \mathbf{w})$, the transition matrices $T$ and $T'$ by SGD strategy, with batchsize of 128, momentum 0.9, weight decay $10^{-3}$, and learning rate $10^{-2}$. For CIFAR-10, the algorithm run 60 epochs and the learning rate is divided by 10 after the 30-$th$ epoch. For CIFAR-100, the algorithm run 80 epochs and the learning rate is divided by 10 after 30-$th$ and 60-$th$ epoch. For Clothing 1M and Food-101N, the backbone we used is ResNet-50 which is pre-trained on ImageNet. We train the classification network $f(\mathbf{x}_i; \mathbf{w})$, the transition matrices $T$ and $T'$ also with SGD strategy, with batchsize of 32, momentum 0.9, weight decay $10^{-3}$, and learning rate $2 \times 10^{-3}$. The algorithm run 80 epochs and the learning rate is divided by 10 every 30 epochs. Before training, we warm up on all noisy data with early stopping technique, where we have trained 10, 10, 1 and 1 epochs on the CIFAR-10, CIFAR-100, Clothing 1M and Food 101N datasets, respectively.

## 3.3 Comparison with state-of-the-art methods

We compare the proposed method with the following representative works: 1) CE, which trains the classification network with the standard cross-entropy loss on the original noisy dataset; 2) GCE [40]; 3) PeerLoss [17]; 4) Co-teaching [6]; 5) Co-teaching++ [38]; 6) T-Revision [33]; 7) Dual-T [37]; VolMinNet [13] which estimates the transition matrix under *sufficiently scattered* assumption.

Table 2: Comparison with state-of-the-art methods on CIFAR-10 and CIFAR-100 datasets. The mean and standard deviation computed over five runs are presented. "Asym-xx%" means the noise rate is xx% and noise type is "Asymmetry".

| Method | Cifar-10 | | | CIFAR-100 | | |
|---|---|---|---|---|---|---|
| | Asym-20% | Asym-40% | Asym-60% | Asym-20% | Asym-40% | Asym-60% |
| CE (baseline) | 84.71±0.21 | 81.26±0.04 | 68.40±1.16 | 52.16±0.37 | 40.21±0.23 | 26.56±0.64 |
| GCE [40] | 89.54±0.21 | 85.95±0.40 | 79.55±0.51 | 65.66±0.73 | 57.34±0.35 | 45.46±0.16 |
| PeerLoss [17] | 88.98±0.15 | 85.61±0.59 | 77.03±0.49 | 57.51±0.05 | 43.95±0.35 | 30.02±0.39 |
| Co-teaching [6] | 85.90±0.38 | 83.09±0.44 | 71.69±0.50 | 57.21±0.37 | 43.76±0.46 | 30.18±0.71 |
| Co-teaching++ [38] | 87.13±0.07 | 84.86±0.36 | 73.50±0.47 | 58.79±0.35 | 45.26±0.41 | 32.02±1.22 |
| T-Revision [33] | 87.99±0.32 | 85.17±0.07 | 72.93±0.27 | 63.94±0.19 | 57.19±1.28 | 42.36±1.09 |
| VolMinNet [13] | 89.62±0.15 | 86.12±0.16 | 74.80±0.15 | 65.91±0.25 | 58.35±0.35 | 42.16±0.94 |
| DualT [37] | 89.36±0.44 | 86.59±0.30 | 78.89±0.99 | 65.76±0.56 | 56.90±0.39 | 44.61±1.20 |
| T-For ($T$) | 89.46±0.21 | 85.74±0.11 | 74.54±0.12 | 65.30±0.01 | 56.31±0.42 | 42.21±0.58 |
| T-Back ($T'$) | 89.97±0.14 | 85.81±0.31 | 73.40±0.81 | 64.56±0.34 | 55.09±0.55 | 41.73±0.73 |
| $T+T'$ | 89.62±0.24 | 86.25±0.03 | 74.80±0.21 | 65.52±0.28 | 57.10±0.20 | 42.72±0.29 |
| Ours | **90.55**±0.03 | **87.29**±0.05 | **82.58**±0.24 | **68.34**±0.24 | **62.64**±0.49 | **50.29**±0.24 |

Table 3: Comparison with state-of-the-art methods on CIFAR-10 and CIFAR-100 datasets. The mean and standard deviation computed over five runs are presented. "Pair-xx%" means the noise rate is xx% and noise type is "Pair".

| Method | Cifar-10 | | | CIFAR-100 | | |
|---|---|---|---|---|---|---|
| | Pair-20% | Pair-40% | Pair-45% | Pair-20% | Pair-40% | Pair-45% |
| CE (baseline) | 86.26±0.41 | 80.24±0.61 | 70.73±1.44 | 52.85±0.35 | 39.66±0.68 | 28.06±0.28 |
| GCE [40] | 89.77±0.17 | 84.11±0.15 | 77.54±0.81 | 65.37±0.29 | 50.91±0.64 | 42.38±0.12 |
| PeerLoss [17] | 90.11±0.03 | 85.52±0.40 | 76.75±2.98 | 60.48±0.08 | 44.55±0.29 | 39.51±0.19 |
| Co-teaching [6] | 86.81±0.31 | 82.29±0.23 | 74.34±0.74 | 59.54±0.14 | 44.97±0.49 | 39.59±0.58 |
| Co-teaching++ [38] | 87.44±0.33 | 83.59±0.33 | 73.96±0.35 | 63.98±0.55 | 46.76±0.73 | 42.60±0.72 |
| T-Revision [33] | 90.90±0.11 | 87.03±0.37 | 75.86±1.16 | 66.37±0.35 | 44.64±0.73 | 38.84±0.47 |
| VolMinNet [13] | 90.99±0.27 | 88.97±0.22 | 74.77±2.15 | 69.70±0.30 | 44.63±0.64 | 39.23±0.17 |
| DualT [37] | 90.18±0.39 | 88.81±0.07 | 77.22±2.08 | 70.07±0.22 | 53.15±0.48 | 42.58±0.54 |
| T-For ($T$) | 90.25±0.40 | 88.40±0.35 | 74.08±1.88 | 69.27±0.14 | 44.65±0.37 | 39.10±0.26 |
| T-Back ($T'$) | 90.03±0.12 | 87.09±0.93 | 73.26±0.89 | 68.61±0.19 | 44.41±0.31 | 38.86±0.44 |
| $T+T'$ | 90.67±0.27 | 89.35±0.49 | 78.62±1.40 | 69.50±0.53 | 44.79±0.65 | 39.16±0.58 |
| Ours | **91.67**±0.27 | **91.36**±0.13 | **91.08**±0.08 | **71.63**±0.39 | **70.87**±0.14 | **69.18**±1.30 |

**Results on the synthetic noisy datasets**. Tables 1, 2 and 3 report the experiment results under three different synthetic noisy types on cifar10 and cifar100 datasets, respectively. Each table compares the proposed method with 8 representative works on one synthetic noisy type. As illustrated in Eq. 5, our overall objective contains three variants as described in the tables: 1) "T-For ($T$)" means that we optimize the classifier just use $\mathcal{L}_1(\mathbf{w}, T)$; 2) "T-Back ($T'$)" means that we optimize the classifier just use $\mathcal{L}_2(\mathbf{w}, T')$; 3) "$T + T'$" means that we optimize the classifier use $\mathcal{L}_1(\mathbf{w}, T) + \mathcal{L}_2(\mathbf{w}, T')$; 4) "Ours" means that we utilize the overall objective as shown in Eq. 5 to optimize the classifier. Compared with the representative methods, the proposed method achieves the best performance on both datasets with three synthetic noise types under three different noise ratios. The evaluation results of the three tables can be summarized as follows:

- Compared with representative methods, the proposed method has significant improvements over these methods, which specifically outperforms the baseline method "CE" and the representative work VolMinNet [13] by a large margin of 16.80% and 7.24% on average.

- As the noise rate increases, the superiority of this method gradually emerges. As shown in these three tables: for symmetric noise, the average advantage of our method over other methods is 4.47%, 6.65% and 10.54% at noise rate of 20%, 40% and 60%, respectively; For asymmetric noise, the average advantage of our method is 4.46%, 6.61% and 10.28% at noise rate of 20%, 40% and 60%, respectively; For pair noise, the average advantage of our method is 4.61%, 15.31% and 22.88% at noise rate of 20%, 40% and 45%, respectively. This indicates that our method can handle difficult cases with high noise rates much better.

Table 4: Classification accuracy (%) on the Clothing1M dataset. (*) indicates that the implementation of the compared method is based on the authors' code.

| Methods | CE (Baseline) | GCE [40] | SL [26] | Co-teaching [6] | JointOpt [23] | $L_{DMI}$ [35] |
|---|---|---|---|---|---|---|
| Accuracy | 68.94 | 69.75 | 71.02 | 69.21 | 72.16 | 72.46 |
| Methods | PTD-R-V [32] | ERL [15] | ForwardT [20] | JoCor [27] | CORES [5] | CAL [41] |
| Accuracy | 71.67 | 72.87 | 69.84 | 70.30 | 73.24 | 74.17 |
| Methods | MEIDTM [4] | VolMinNet* [13] | Ours | DivideMix* [11] | DivideMix+VolMinNet | DivideMix+Ours |
| Accuracy | 73.05 | 69.82 | 70.73 | 74.67 | 74.83 | **75.12** |

Table 5: Classification accuracy (%) on the Food-101N dataset. (*) indicates that the implementation of the compared method is based on the authors' code.

| Methods | CE (Baseline) | CleanNet$_{WH}$ [10] | CleanNet$_{WS}$ [10] | DeepSelf [7] | NoiseResist [14] | VolMinNet* [13] |
|---|---|---|---|---|---|---|
| Accuracy | 81.44 | 83.47 | 83.95 | 85.11 | 84.70 | 83.04 |
| Methods | DivideMix* [11] | Ours | DivideMix+T-For ($T$) | DivideMix+T-Back ($T'$) | DivideMix+VolMinNet | DivideMix+Ours |
| Accuracy | 84.39 | 83.71 | 85.07 | 84.83 | 85.07 | **86.11** |

- By comparing the four variants of our method, it clearly shows that integrating the backward transition matrix and cycle-consistency regularization can greatly help to improve the baseline performances step by step.

**Results on the real-world datasets.**. Tables 4 and 5 compare the proposed method with representative works on Clothing1M and Food-101N datasets, respectively. We can clearly see that our methods outperforms the baseline method "CE" and VolMinNet [13] by a margin of 1.79% and 0.91% on Clothing1M dataset, and 2.27% and 0.67% on Food-101N dataset. What's more, since the proposed method can be used as a plug-and-play module, we integrate this module into the representative work DivideMix [11], and denoted as "DivideMix+Ours". To further illustrate its effectiveness, we also combine DivideMix with VolMinNet as another baseline, denoted as "DivideMix+VolMinNet". We can clearly see that our method achieves superior performances over all these methods.

### 3.4   Ablation Study

As shown in Figure 2 (a) to (l), we also show the estimation error of the transition matrix $T$ of our method during the model training, with different noise types and different noise rates, on the CIFAR-10 and CIFAR-100 datasets, to evaluate the estimated transition matrix $T$. The estimation error is measured by the $l_1$ norm between the ground-truth transition matrix and the estimated transition matrix. It shows that the proposed method always outperforms previous algorithms on $T$ estimation. To explore the effect of hyper-parameter $\lambda$ on model performance, we conducted experiments with different $\lambda$ under different noise types and noise rates on CIFAR-10 and cifar-100 datasets, as shown in Figure 2 (m) to (p). Each experiment run five times. We summarize Figure 2 (m) to (p) as follows: 1) The smaller the $T$ estimation error is, the higher the test accuracy will be; 2) When $\lambda = 0.3$, the estimation error is the smallest and we get the best performances. Based on this observation, we set $\lambda$ as 0.3 in all our experiments.

## 4   Related Work

Based on the statistical consistency of the learned classifier, we roughly divide existing label-noise learning methods into two categories: algorithms with statistically inconsistent classifiers (i.e., heuristic algorithms), and the algorithms with statistically consistent classifiers.

**The statistically inconsistent algorithms** usually explore some heuristics to reduce the side-effect of the noisy labels. The representative works include: 1) the data cleaning methods which specially design some strategies to select reliable examples [18, 5]; 2) the "label correction" methods which aim to improve the label quality during model training [23, 21]; 3) the semi-supervised learning methods which treat the unreliable examples as unlabeled data and then adopt some self-supervised training technics for robust feature representation learning [11]; 4) some other classic machine learning techniques [19, 20], such as soft label based methods [29], early stop tricks to avoid over-fitting [2, 31]. Although these methods empirically work well without explicitly modeling the label noise distribution, we can not theoretically guarantee the consistency of the learned classifiers [6].

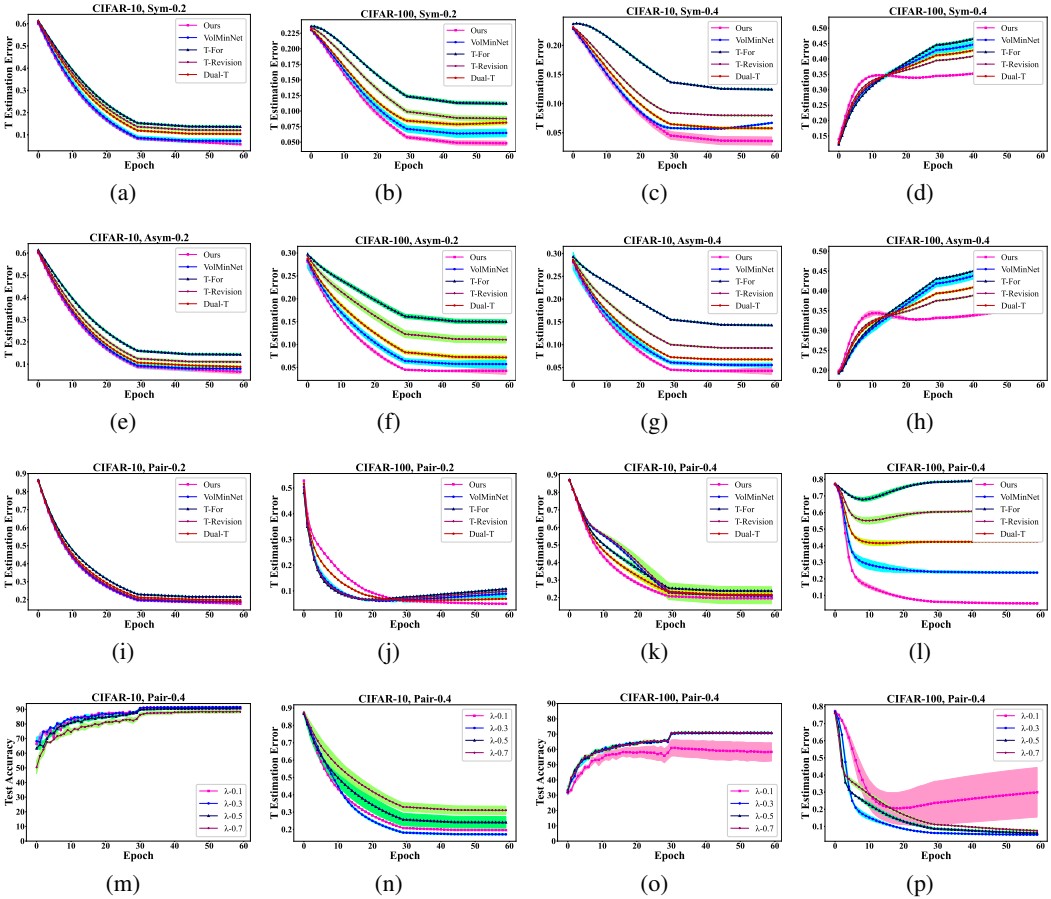

Figure 2: (a) to (l) compare the estimation error of $T$ between our method and other transition matrix based methods with different noise types and noise ratios on two datasets. (m) to (p) show classification accuracy and $T$ estimation error with various values of $\lambda$ on two datasets.

**The statistically consistent algorithms** aim to learn consistent classifier with the noisy data, which will asymptotically converge to the optimal classifier obtained on its corresponding clean risk[37]. These methods are primarily designed based on the "loss correction" strategy. Among them, the transition matrix $T(\mathbf{x})$ has been used to modify the loss function to build consistent classifiers. Specifically, the anchor point assumption has been widely adopted to estimate the transition matrix [16, 33], and some methods directly assume that the anchor points have already been given. However, this assumption sometimes could be too strong in real application. When no anchor point is given in the dataset, these algorithms can not be guaranteed statistically consistent. Then, to remove this strong dependence on anchor points, some methods propose the slack variable trick [25], which just use the data points with high noisy class-posterior probabilities to estimate the transition matrix.

Afterwards, a series of assumptions [4, 32, 13] have been proposed to efficiently estimate the transition matrix. For example, Dual-T estimator [37] introduces the intermediate class to factorize the original transition matrix into the product of two easy-to-estimate transition matrices. ExtendedT [30] proposed to extend the traditional transition matrix to be able to model mixed close-set and open-set label noise. VolMinNet [13] try to consistently estimate the transition matrix under the *sufficiently scattered* assumption, which empirically incorporates the minimum volume constraint of $T$ into the label-noise learning. However, these methods relies heavily on the estimated inaccurate noisy class-posterior probability, which could lead the transition matrix to be poorly estimated.

## 5    Conclusion and Limitation

Estimating the transition matrix $T$ plays an important role in building statistically consistent classifier for label-noise learning. To address the bottleneck of how to reduce the side-effects of the large

estimation error of the noisy class posterior on transition matrix estimation, this paper proposes to estimate $T$ under a forward-backward cycle-consistency regularization, of which we could greatly reduce the dependency of estimating $T$ on the estimated noisy class posterior. We also show that the proposed method helps to minimize the *volume* of $T$ without directly exploiting the estimated noisy class posterior, which encourages the estimated transition matrix $T$ to converge to its optimal solution. Extensive experimental results consistently justify the effectiveness of the proposed method, from both of the superior classification accuracy and the estimation error of the learned transition matrix. **Limitation.** One major limitation in this study is that our method is constrained to work well under the *sufficiently scattered* assumption. When this assumption is violated in some cases, the statistical consistency of the estimated transition matrix and classifier would not be guaranteed. In the future, we will make deep theoretical analysis on the backward transition matrix and estimated clean class posterior probability.

**Acknowledgements:** This work was supported in part by the National Natural Science Foundation of China under Grant 62176198, 61922066 and 62106184, in part by the National Key Research and Development Program of China under Grant 2018AAA0103202, in part by the Technology Innovation Leading Program of Shaanxi under Grant 2022QFY01-15, in part by Open Research Projects of Zhejiang Lab under Grant 2021KG0AB01. Tongliang Liu was partially supported by Australian Research Council Projects DP180103424, DE-190101473, IC-190100031, DP-220102121, and FT-220100318. Bo Han was supported by the RGC Early Career Scheme No. 22200720, NSFC Young Scientists Fund No. 62006202, and Guangdong Basic and Applied Basic Research Foundation No. 2022A1515011652.

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

# A    Appendix

Optionally include extra information (complete proofs, additional experiments and plots) in the appendix. This section will often be part of the supplemental material.

