# Supplementary Material

## 1 Definition of Noise Models

On CIFAR-10 and CIFAR-100, following the traditional methods [3], we manually corrupt the training set according to the ground-truth transition matrices $T$, where $T_{ij} = P(\tilde{y} = j | y = i)$ given that noisy label $\tilde{y}$ is flipped from clean label $y$.

As described in [1], the Noise transition matrix supposes that the observed noisy label $\tilde{y}$ is drawn independently from a corrupted distribution $P(X, \tilde{Y})$, where features are intact. Meanwhile, there exists a corruption process, transition from the latent clean label $y$ to the observed noisy label $\tilde{y}$. Such a corruption process can be approximately modeled via noise transition matrix $T$, where $T_{ij} = P(\tilde{y} = j | y = i)$.

Specifically, we conduct experiments using three commonly used noisy types: 1)Symmetry flipping [4]; 2) Asymmetry flipping [4]; 3) Pair flipping [2].

### 1.1 The transition matrix $T$ of the Symmetry Flipping noise type

In the following, $\varepsilon$ is the noise rate, $C$ is number of classes, and the transition matrix $T \in \mathbb{R}^{C \times C}$.

$$T = \begin{bmatrix} 1-\varepsilon & \frac{\varepsilon}{C-1} & \cdots & \frac{\varepsilon}{C-1} & \frac{\varepsilon}{C-1} \\ \frac{\varepsilon}{C-1} & 1-\varepsilon & \cdots & \frac{\varepsilon}{C-1} & \frac{\varepsilon}{C-1} \\ \vdots & \vdots & \ddots & \vdots & \vdots \\ \frac{\varepsilon}{C-1} & \frac{\varepsilon}{C-1} & \cdots & 1-\varepsilon & \frac{\varepsilon}{C-1} \\ \frac{\varepsilon}{C-1} & \frac{\varepsilon}{C-1} & \cdots & \frac{\varepsilon}{C-1} & 1-\varepsilon \end{bmatrix} \tag{1}$$

### 1.2 The transition matrix $T$ of the Pair Flipping noise type

In the following, $\varepsilon$ is the noise rate, $C$ is number of classes, and the transition matrix $T \in \mathbb{R}^{C \times C}$.

$$T = \begin{bmatrix} 1-\varepsilon & \varepsilon & 0 & \cdots & 0 \\ 0 & 1-\varepsilon & \varepsilon & \cdots & 0 \\ \vdots & \vdots & \ddots & \vdots & \vdots \\ 0 & 0 & \cdots & 1-\varepsilon & \varepsilon \\ \varepsilon & 0 & \cdots & 0 & 1-\varepsilon \end{bmatrix} \tag{2}$$

### 1.3 The transition matrix $T$ of the Asymmetry Flipping noise type

The asymmetric label noise is designed to mimic some structure of the real mistakes for similar classes: $TRUCK \longrightarrow AUTOMOBILE$, $BIRD \longrightarrow AIRPLANE$, $DEER \longrightarrow HORSE$, $CAT \longleftrightarrow DOG$ [4]. Label transition matrix are parameterized by $\epsilon \in [0, 1]$ such that the true class and wrong class have probability of $1 - \epsilon$ and $\epsilon$, respectively. An example of $T$ used for CIFAR-10

Preprint. Under review.

dataset with $\epsilon = 0.7$ is shown as follows.

$$T = \begin{bmatrix} 1 & 0 & 0 & 0 & 0 & 0 & 0 & 0 & 0 & 0 \\ 0 & 1 & 0 & 0 & 0 & 0 & 0 & 0 & 0 & 0 \\ 0 & 0 & 0.3 & 0 & 0 & 0 & 0 & 0.7 & 0 & 0 \\ 0 & 0 & 0 & 0.3 & 0 & 0 & 0 & 0 & 0.7 & 0 \\ 0 & 0 & 0 & 0 & 1 & 0 & 0 & 0 & 0 & 0 \\ 0 & 0 & 0 & 0 & 0 & 0.3 & 0.7 & 0 & 0 & 0 \\ 0 & 0 & 0 & 0 & 0 & 0.7 & 0.3 & 0 & 0 & 0 \\ 0 & 0.7 & 0 & 0 & 0 & 0 & 0 & 0.3 & 0 & 0 \\ 0 & 0 & 0 & 0 & 0 & 0 & 0 & 0 & 1 & 0 \\ 0 & 0 & 0 & 0 & 0 & 0 & 0 & 0 & 0 & 1 \end{bmatrix} \tag{3}$$