# OpenReview forum: "Class-Dependent Label-Noise Learning with Cycle-Consistency Regularization"
_NeurIPS.cc/2022/Conference — NeurIPS 2022 Accept_

### Official Review · Reviewer_4D9u · 2022-07-08

**Rating:** 7
**Confidence:** 5
**Soundness:** 4 excellent
**Presentation:** 3 good
**Contribution:** 3 good

**Summary:**

This paper focuses on the class-dependent noisy labels. To solve this, the authors proposes a cycle-consistency regularization on the estimation of the transition matrix for learning with class-dependent noisy-labels. The proposed method could encourage the estimated transition matrix to converge to its optimal solution, without explicitly estimating the noisy class posterior probability. Therefore, it could help to build a better statistically consistent classifier. Experimental results on several datasets show the effectiveness of this method, on reducing the estimation error of T and boosting the classification performance.

**Questions:**

See the weaknesses above.

**Limitations:**

The authors have discussed the limitations of this paper, and there is no negative societal impact.

**Strengths And Weaknesses:**

Strengths:
1. Different from the previous representative work VolMinNet, this paper proposes a new strategy to optimize the transition matrix T. The proposed method successfully addresses the problem of how to reduce the side-effects of the inaccurate noisy class posterior on estimating the transition matrix T.
2. The author conduct experiment on several datasets to show the effectiveness of the proposed method. Moreover, the theoretical analysis is also provided to demonstrate the effectiveness of the algorithm.

Weaknesses:
1. In the overall objective function shown in Eq. (4), I believe we could also obtain the experiment results by just optimizing the backward transition matrix T^b. That is to say, we could just minimize Eq. (3) to obtain one intermediate results. However, this paper does not list this result for experimental ablation study.
2. In line 43-44, why anchor point assumption is a special case of the sufficiently scattered assumption?
3. The author claimed that the proposed method can be used as a plug-and-play module and integrate it into the DivideMix. The authors should clearly define the experiment settings and how they are combined.

---

> ### Author Response · Authors · 2022-08-02
> **Response to Reviewer's (Reviewer 4D9u) comments**
>
> 1、Ablation study on just using Eq.(3) to optimize the classification model.
>
> The overall objective function as shown in Eq.(5) contains three items, namely: T-Forward transition matrix module ($T$-For), T-Backward Transition matrix ($T^{b}$), their combination ($T+T^{b}$), and our final proposed method (Ours). We have done detailed ablation study to reveal how each item contributes to the overall method and performance improvement, which includes all the intermediate results (Eq.(2),Eq.(3) and Eq.(4)). All the experiment results on two synthetic datasets and two real-world datasets are illustrated in the following table. We can clear see that just using T-Forward and T-Backward transition matrix could obtain comparable experiment results, where the T-Forward is slightly better. When we combine above items step-by-step, much performance improvement could be obtained.
>
> | Dataset  | Cifar-10 |
> | Method  | Sym-20 | Sym-40| Sym-60 | Asym-20 |  Asym-40 | Asym-60 | Pair-20|  Pair-40 | Pair-60 |
> | :-----------: | ----: | ----: | :----: | ----: | ----: | :----: | ----: | ----: | :----: |
> | T-For ($T$) | 89.53$\pm$0.11 | 85.38$\pm$0.13 | 73.01$\pm$0.54 | 89.46$\pm$0.21 | 85.74$\pm$0.11| 74.54$\pm$0.12 |  90.25$\pm$0.40 | 88.40$\pm$0.35 | 74.08$\pm$1.88 |
> | T-Back ($T^{b}$) | 88.40$\pm$0.12 | 84.97$\pm$0.16 | 73.12$\pm$0.79 | 88.97$\pm$0.14 | 85.81$\pm$0.31 | 73.40$\pm$0.81 |   90.03$\pm$0.12 | 87.09$\pm$0.93 | 73.26$\pm$0.89 |
> | $T+T^{b}$ | 89.64$\pm$0.16 | 85.47$\pm$0.32 | 73.39$\pm$0.40 | 89.62$\pm$0.24 | 86.25$\pm$0.03 | 74.80$\pm$0.21 |   90.67$\pm$0.27 | 89.35$\pm$0.49 | 78.62$\pm$1.40 |
> | ours | 90.44$\pm$0.19 | 87.30$\pm$0.25| 81.01$\pm$0.25 | 90.55$\pm$0.03 |87.29$\pm$0.05| 82.58$\pm$0.24 |   91.36$\pm$0.13 | 91.08$\pm$0.08 | 71.63$\pm$0.39 |
>
>
> | Dataset  | Cifar-100 |
> | Method  | Sym-20 | Sym-40| Sym-60 | Asym-20 |  Asym-40 | Asym-60 | Pair-20|  Pair-40 | Pair-45 |
> | :-----------: | ----: | ----: | :----: | ----: | ----: | :----: | ----: | ----: | :----: |
> | T-For ($T$) | 64.23$\pm$0.64 | 56.02$\pm$0.39 | 40.89$\pm$0.37 | 65.30$\pm$0.01 | 56.31$\pm$0.42 | 42.21$\pm$0.58 | 69.27$\pm$0.14 | 44.65$\pm$0.37 | 39.10$\pm$0.26 |
> | T-Back ($T^{b}$) | 63.39$\pm$0.62 | 54.96$\pm$0.43 | 41.15$\pm$0.82 | 64.56$\pm$0.34 | 55.09$\pm$0.55 | 41.73$\pm$0.73 | 68.61$\pm$0.19 | 44.41$\pm$0.31 | 38.86$\pm$0.44 |
> | $T+T^{b}$ | 64.95$\pm$0.91 | 56.36$\pm$0.51 | 41.94$\pm$0.43 | 65.52$\pm$0.28 | 57.10$\pm$0.20 | 42.72$\pm$0.29 | 69.50$\pm$0.53 | 44.79$\pm$0.65 | 39.16$\pm$0.58 |
> | ours | 67.74$\pm$0.17 | 61.71$\pm$0.20 | 49.30$\pm$0.82 | 68.34$\pm$0.24 | 62.64$\pm$0.49 | 50.29$\pm$0.24 | 71.63$\pm$0.39 | 70.87$\pm$0.14 | 69.18$\pm$1.30 |
>
>
> 2、In line 43-44, why anchor point assumption is a special case of the sufficiently scattered assumption?
>
> As described in the previous representative work VolMinNet[14], the anchor-point assumption is a sufficient but not necessary condition for the sufficiently scattered assumption when the number of classes $C>2$. The proof of the above proposition can be found in VolMinNet[14]. To briefly answer this question, we first illustrate the definition of the anchor point assumption. For each class $k\in {1,...,C}$, if there exists an instance $\textbf{x}^k\in X$ with $P(Y=k|X=x^{k}) = 1$, then we define these examples are anchor points. Under the anchor point assumption, the transition matrix based label noise learning method focuses on finding anchor points for each class. Intuitively, if the anchor point assumption is satisfied, then there exists a matrix $\textbf{V}=[P(Y|X=\textbf{x}^1),...,P(Y|X=\textbf{x}^C)]=\textbf{I}$, where $\textbf{x}^1,...,\textbf{x}^C$ are anchor points for different classes and $\textbf{I}$ is the identify matrix. As illustrated in VolMinNet[14], the convex cone formed by the columns of $V$ denoted as $cone\{V\}$. We can clearly see that $cone\{V\}=cone\{\textbf{I}\}$, and the $cone\{V\}$ can only be enclosed by the convex cone of permutation matrices. This shows that sufficiently scattered assumption is satisfied. Therefore, anchor point assumption is a special case of the sufficiently scattered assumption, but vice versa.
>
> 3、The authors should clearly define the experiment settings and how they are combined.
>
> DivideMix is one representative semi-supervised method which integrated several technics to help improve the model performance. First, it uses a a Gaussian mixture model to model the loss distribution for each sample, then it dynamically splits the training data into the clean labeled subset and unlabeled subset with noisy samples. And finally adopt the semi-supervised method to train the model with these labeled and unlabeled data. However, there still contain many noisy samples in the filtered clean subset that are considered to be clean. Therefore, when training model on these selected clean subset with the supervised method, i.e., cross-entropy loss, we could also integrate our proposed transition matrix estimation module into the DivideMix framework.

---

> > ### Comment · Reviewer_4D9u · 2022-08-08
> > **Thanks for the reply**
> >
> > Thanks for the detailed reply and new results. The authors have addressed my concerns well and I would keep my recommendation.

---

### Official Review · Reviewer_f3fT · 2022-07-09

**Rating:** 7
**Confidence:** 5
**Soundness:** 3 good
**Presentation:** 4 excellent
**Contribution:** 4 excellent

**Summary:**

This paper proposes a label-noise learning algorithm through estimating the transition matrix T, to build a statistically consistent classifier. Specifically, the proposed algorithm tries to estimate the transition matrix under the forward-backward cycle-consistency regularization, which helps to minimize the volume of the transition matrix indirectly without exploiting the estimated noisy class posterior. Theoretical analysis and experimental results justify the effectiveness of the proposed method.

**Questions:**

see the weaknesses above

**Limitations:**

YES

**Strengths And Weaknesses:**

Strengths:
1)	The proposed method is technical sound and novel. It develops an algorithm to estimate the transition matrix under the sufficiently scattered assumption, by incorporating the proposed cycle-consistency regularization. This method can reduce the dependency of estimating the transition matrix T on the inaccurately estimated noisy class posterior.
2)	Detailed theoretical analysis illustrates that the cycle-consistency regularization helps to minimize the volume of the transition matrix T, which encourages the estimated T to converge to its optimal solution.
3)	The experiment is sufficient. They illustrate the effectiveness of the proposed method from both of the classification performance and the T estimation error, on both the synthetic and real-world noisy datasets.
Weaknesses:
1)	Since this method needs a forward-backward training process, does it require more time to train the network?  How many extra parameters are introduced in the newly proposed method compared with the previously proposed method VolMinNet[13]?
2)	This paper states that it tries to estimate the transition matrix under the sufficiently scattered assumption, what’s the difference between this assumption and the previous anchor point assumption?

---

> ### Author Response · Authors · 2022-08-02
> **Response to Reviewer's comments**
>
> 1、 Since this method needs a forward-backward training process, does it require more time to train the network? How many extra  arameters are introduced in the newly proposed method compared with the previously proposed method VolMinNet[13]?
>
> Compared with the previous representative work VolMinNet[13], our proposed method does not need to directly compute the volume of the transition matrix $T$, i.e., vol($T$). In practice, vol($T$) denotes a measure that is related or proportional to the volume of the simplex formed by the column of $T$. Given a square matrix $T$, the VolMinNet method adopts the determinant of the matrix as the the volume measurement, i.e., vol$(T)=$ det$(T)$, where det denotes the determinant. Clearly, the VolMinNet needs to compute the determinant of the transition matrix $T$ during model training. While our proposed method needs to optimize the backward transition matrix $T^{b}$, through minimizing another two cross-entropy loss, and it will also introduce extra $C\times C$ parameters, where $C$ is the number of classes. However, the introduced $C\times C$ extra parameters in our method are relatively very small compared with the overall parameters of the whole model. Also, both the forward and backward transition matrix $T$ and $T^{b}$  can be optimized in an end-to-end manner simultaneously. Therefore, our method requires almost the same training time as the compared method VolMinNet in practice, just introduces another $C\times C$ extra parameters.
>
> 2)This paper states that it tries to estimate the transition matrix under the sufficiently scattered assumption, what’s the difference between this assumption and the previous anchor point assumption?
>
> The anchor point assumption assumes that there contain some instances belonging to a specific class almost surely. The definition of anchor point assumption can be described as follows. For each class $k\in {1,...,C}$, if there exists an instance $\textbf{x}^k\in X$ with $P(Y=k|X=x^{k}) = 1$, then we define these examples are anchor points. Under the anchor point assumption, the transition matrix based label noise learning method focuses on finding anchor points for each class. While for the sufficiently scattered assumption, the definition can be clearly stated according to VolMinNet. The difference or relationship between them is that, the anchor-point assumption is a sufficient but not necessary condition for the sufficiently scattered assumption when $C>2$. That is to say, the anchor-point assumption is a special case of the sufficiently scattered assumption, and the  sufficiently scattered assumption is a mild assumption to the anchor-point assumption, which can be theoretically proved. Therefore, the proposed method under the sufficiently scattered assumption, could deal with more general and complex noise models.

---

> > ### Comment · Reviewer_f3fT · 2022-08-10
> > **post rebuttal**
> >
> > The response well addresses my concern and I would keep my score.

---

### Official Review · Reviewer_zugN · 2022-07-11

**Rating:** 8
**Confidence:** 4
**Soundness:** 3 good
**Presentation:** 3 good
**Contribution:** 3 good

**Summary:**

This paper proposes the idea of using the forward-backward cycle-consistency regularization to estimate the transition matrix T. This method addresses a long-standing problem of estimating the transition matrix depending on the inaccurately estimated noisy class posterior probability. Theoretical analysis and experimental results both show that the proposed method is superior to the compared methods. Particularly, it seems that this paper firstly proposes to estimate the transition matrix T in a backward manner.

**Questions:**

Besides answering the above listed weaknesses, I also curious about the following question: Why not directly use T^-1 instead of the learned T^b to optimize the transition matrix?

**Limitations:**

Yes.

**Strengths And Weaknesses:**

Strengths:
1) Novelty: The main contribution of this work is the idea of estimating the transition matrix T with the forward-backward cycle-consistency regularization, under the sufficiently scattered assumption. Maybe, this is the first to estimate the transition matrix T bidirectionally.
2) Writing: The paper looks good and well-written. The introduction and the algorithm part are easy to follow.
3) Experiments: The proposed method achieves better results in both the synthetic and real-world datasets, and provides detailed ablation study.

Weaknesses:
1) It needs some model complexity analysis, especially for the training process of the backward and forward transition matrix.
2) Some implementation details are missing. How do you combine the proposed transition matrix estimating approach with the traditional DivideMix[11] algorithm, as shown in Table 4 and Table 5 ?

---

> ### Author Response · Authors · 2022-08-01
> **Response to Reviewer2's comments**
>
> 1、It needs some model complexity analysis, especially for the training process of the backward and forward transition matrix.
>
> Compared with the previous representative work VolMinNet[13], our proposed method does not need to directly compute the volume of the transition matrix $T$, i.e., vol($T$), during model training.  But  the proposed method needs to optimize the backward transition matrix $T^{b}$, through minimizing another two cross-entropy loss, and it will also introduce extra $C\times C$ parameters, where $C$ is the number of classes. However, the introduced $C\times C$ extra parameters in our method are relatively very small compared with the overall parameters of the whole model. Also, both the forward and backward transition matrix $T$ and $T^{b}$  can be optimized in an end-to-end manner simultaneously. Therefore, our method requires almost the same training time as the compared method VolMinNet in practice, just introduces another $C\times C$ extra parameters.
>
>
> 2、Some implementation details are missing. How do you combine the proposed transition matrix estimating approach with the traditional DivideMix[11] algorithm.
>
> DivideMix uses a Gaussian mixture model to model the loss distribution for each sample, dynamically splits the training data into the clean labeled subset and unlabeled subset with noisy samples. Then it adopts the semi-supervised method to train the model with these labeled and unlabeled data. However, there still contain many noisy samples in the filtered clean subset that are considered to be clean. Therefore, when training model on these selected clean subset with the supervised method, i.e., cross-entropy loss, we could also integrate our proposed transition matrix estimation module into the DivideMix framework. Specifically, we utilize the proposed method to further model the label noise in the filtered clean subset
>
>
> 3、Besides answering the above listed weaknesses, I also curious about the following question: Why not directly use $T^{-1}$ instead of the learned $T^b$ to optimize the transition matrix?
>
> Different from the traditional method that always estimate the transition matrix $T$ through minimizing the cross-entropy loss between the noisy class-posterior probability $P(\bar{\textbf{Y}}|X)$ and the given noisy label $\bar{y}$, under specific constraints. We also propose to estimate the backward transition matrix $T^{b}$ simultaneously, to act as $T^{-1}$. Then we can also build the consistency regularization. However, since the transition matrix is to model the noisy data generation process, each element in the transition matrix has its physical meaning. Specifically, we always maintain the forward and backward transition matrix ($T$ and $T^b$) be diagonally dominant column stochastic matrix. Therefore, directly computing $T^{-1}$ cannot satisfy this constraint, and the backward transition matrix $T^b$ will be different from $T^{-1}$. Most important, the newly computed backward transition matrix is worked as the regularization term aiming to maximize the volume of the clean class posterior probability. Also, through the consistency regularization term, we could make full use of the invertible relationship between these two matrices $T$ and $T^b$. Finally, it could encourage the estimated transition matrix to converge to the optimal solution.

---

### Official Review · Reviewer_3zVp · 2022-07-11

**Rating:** 3
**Confidence:** 4
**Soundness:** 1 poor
**Presentation:** 1 poor
**Contribution:** 2 fair

**Summary:**

The paper proposes a method to achieve good prediction accuracy under label noise, by estimating the transition matrix from clean conditional probabilities, to noisy conditional probabilities. The paper presents experiments on synthetic noise models with a known transition matrix and natural noise models.


**Questions:**

- What is the motivation for the method proposed in section 2.2? For instance, why is it necessary to learn the second matrix $T^b$?

- What happens if one only minimizes the second term in equation 5? Some of the experiments imply good performance and it would be interesting to understand what is the reason behind the method’s success in those scenarios. A more thorough ablation study would likely reveal what makes the method work well and what is perhaps unnecessary and can be stripped away from the loss.
It is difficult to assess the performance of the method because crucial details about the experimental setup are missing. What are the exact noise models? How many classes are assumed to suffer from label noise in the symmetric/asymetric/pair model? What was the procedure used for hyperparameter tuning for the baselines and for the proposed method?

- The method seems to not perform particularly well on datasets with natural noise. Why are only some and not all baselines presented in tables 4 and 5? What are the confidence intervals for the values in tables 4 and 5? What are possible reasons why the proposed method performs worse on natural noise models compared to the synthetic noise instances? I suggest adding to the experimental comparison more settings with noise models other than the ones considered in Tables 1-3, e.g. random flips for all classes with equal probability, other natural label noise datasets (e.g. OpenImages, MS-COCO with the noisy annotations, the dataset of https://github.com/zhongyy/Unequal-Training-for-Deep-Face-Recognition-with-Long-Tailed-Noisy-Data etc).

- How does a simple baseline that uses early stopping regularization (see e.g. https://arxiv.org/pdf/1910.09055.pdf and http://proceedings.mlr.press/v108/li20j/li20j.pdf) compare to the proposed method?


**Limitations:**

It is unclear from the results reported in the tables (in particular Tables 2 and 3) when the proposed method breaks and what exactly are the assumptions that it requires to perform well. Instead of selecting the 3 fixed noise rates reported in the paper, I suggest a plot in which the noise rate is varied on the Ox axis, while the Oy axis indicates the test accuracy. See also the discussion regarding other noise models in the Questions section.


**Strengths And Weaknesses:**

Strengths:

- Some of the experimental results are promising, especially on the synthetic noise models (Tables 1-3 and Figure 2).

Weaknesses:

- The paper could benefit from some changes regarding its style, which is atypical and unfortunately hurts clarity quite significantly. Here are just a few examples of things that could be improved. The introduction contains heavy notation that is not necessary at this point. The introduction even contains the proof of the “theoretical result”, as the paper notes on line 188. The paper is also not self-contained, and important concepts are not defined at all, e.g. the sufficiently scattered assumption or the anchor point assumption, the sym, asym or pair noise models, the baselines etc. In addition, the writing is at times too colloquial (e.g. “we creatively propose”, “what’s more” etc).  There are also numerous typos and grammar mistakes (line 39, 53 etc).

- The paper does not explain how the authors arrived at the procedure described in section 2.2. (see also the Questions section). There is also very little discussion on the impact of the 3 terms in the loss, or why the loss in equation 5 is particularly well suited for this problem.

- It is not possible to review the soundness of the experimental evaluation since too many details are currently missing from the manuscript.

---

> ### Author Response · Authors · 2022-08-01
> **Response to Reviewer's (Review3zVp) comments (Part 1)**
>
> 1、Motivation of the proposed method: 1) Current state-of-the-art consistent estimator for transition matrix has been developed via incorporating the minimum volume constraint of $T$ into label-noise learning. However, computing the volume of $T$ heavily relies on the inaccurately estimated noisy class posterior $P(\bar{\textbf{Y}}|X=\textbf{x})$, which could lead the transition matrix $T$ to be poorly estimated. Instead, our method theoretically proves that minimizing the volume of $T$ is equal to maximizing the volume constraint of the clean class posterior $P(\textbf{Y}|X=\textbf{x})$, to reduce the side-effects of the inaccurate estimated noisy class posterior. To obtain the clean class posterior with maximum volume, we propose to regularize it through the backward transition matrix $T^{b}$ and the given noisy label $\bar{y}$.  2) Different from the traditional method, we also propose to estimate the backward transition matrix $T^{b}$ simultaneously, to act as $T^{-1}$. When we obtain the forward and backward transition matrices $T$ and $T^{b}$, we could build the ``indirect’’ cycle-consistency through minimizing the approximation error between $P(\textbf{Y}|X=\textbf{x})$ and $T^{b}(TP(\textbf{Y}|X=\textbf{x}))$, which could make full use of the invertible relationship between these two matrices $T$ and $T^{b}$ indirectly. Based on the above two reasons, it is very necessary to learn the backward transition matrix $T^b$, though which it could encourage the estimated $T$ to converge to its optimal solution.
>
> 2、Ablation study on just minimizing the second term in Eq.5.
>
> We have done detailed ablation study to reveal how each item contributes to the overall method and performance improvement, which includes all the intermediate results (Eq.(2),Eq.(3) and Eq.(4)). All the experiment results on two synthetic datasets and two real-world datasets are illustrated in the following table. We can clear see that just using T-Forward and T-Backward transition matrix could obtain comparable experiment results, where the T-Forward is slightly better. When we combine above items step-by-step, much performance improvement could be obtained.
>
> | Dataset  | Cifar-10 |
> | Method  | Sym-20 | Sym-40| Sym-60 | Asym-20 |  Asym-40 | Asym-60 | Pair-20|  Pair-40 | Pair-60 |
> | :-----------: | ----: | ----: | :----: | ----: | ----: | :----: | ----: | ----: | :----: |
> | T-For ($T$) | 89.53$\pm$0.11 | 85.38$\pm$0.13 | 73.01$\pm$0.54 | 89.46$\pm$0.21 | 85.74$\pm$0.11| 74.54$\pm$0.12 |  90.25$\pm$0.40 | 88.40$\pm$0.35 | 74.08$\pm$1.88 |
> | T-Back ($T^{b}$) | 88.40$\pm$0.12 | 84.97$\pm$0.16 | 73.12$\pm$0.79 | 88.97$\pm$0.14 | 85.81$\pm$0.31 | 73.40$\pm$0.81 |   90.03$\pm$0.12 | 87.09$\pm$0.93 | 73.26$\pm$0.89 |
> | $T+T^{b}$ | 89.64$\pm$0.16 | 85.47$\pm$0.32 | 73.39$\pm$0.40 | 89.62$\pm$0.24 | 86.25$\pm$0.03 | 74.80$\pm$0.21 |   90.67$\pm$0.27 | 89.35$\pm$0.49 | 78.62$\pm$1.40 |
> | ours | 90.44$\pm$0.19 | 87.30$\pm$0.25| 81.01$\pm$0.25 | 90.55$\pm$0.03 |87.29$\pm$0.05| 82.58$\pm$0.24 |   91.36$\pm$0.13 | 91.08$\pm$0.08 | 71.63$\pm$0.39 |
>
> | Dataset  | Cifar-100 |
> | Method  | Sym-20 | Sym-40| Sym-60 | Asym-20 |  Asym-40 | Asym-60 | Pair-20|  Pair-40 | Pair-45 |
> | :-----------: | ----: | ----: | :----: | ----: | ----: | :----: | ----: | ----: | :----: |
> | T-For ($T$) | 64.23$\pm$0.64 | 56.02$\pm$0.39 | 40.89$\pm$0.37 | 65.30$\pm$0.01 | 56.31$\pm$0.42 | 42.21$\pm$0.58 | 69.27$\pm$0.14 | 44.65$\pm$0.37 | 39.10$\pm$0.26 |
> | T-Back ($T^{b}$) | 63.39$\pm$0.62 | 54.96$\pm$0.43 | 41.15$\pm$0.82 | 64.56$\pm$0.34 | 55.09$\pm$0.55 | 41.73$\pm$0.73 | 68.61$\pm$0.19 | 44.41$\pm$0.31 | 38.86$\pm$0.44 |
> | $T+T^{b}$ | 64.95$\pm$0.91 | 56.36$\pm$0.51 | 41.94$\pm$0.43 | 65.52$\pm$0.28 | 57.10$\pm$0.20 | 42.72$\pm$0.29 | 69.50$\pm$0.53 | 44.79$\pm$0.65 | 39.16$\pm$0.58 |
> | ours | 67.74$\pm$0.17 | 61.71$\pm$0.20 | 49.30$\pm$0.82 | 68.34$\pm$0.24 | 62.64$\pm$0.49 | 50.29$\pm$0.24 | 71.63$\pm$0.39 | 70.87$\pm$0.14 | 69.18$\pm$1.30 |
>
>
> | Dataset  | Cloting1M | Food-101N|
> | Method  | Cloting1M | Food-101N|
> | :-----------: | ----: | :----: |
> | DivideMix | 74.58 | 84.37 |
> | DivideMix+T-For ($T$) | 74.83 | 85.07 |
> | DivideMix+T-Back ($T^{b}$) | 74.75 | 84.83 |
> | DivideMix+ours | 75.12$\pm$0.05 | 86.11$\pm$0.03 |
>
>
> 3、What are possible reasons why the proposed method performs worse on natural noise models compared to the synthetic noise instances?：
>
> The natural noise models are always instance-dependent label noise, which are more complex than the synthetic label noise with predefined noise label distribution. Besides, there are also some open-set labels in the real-world datasets with noisy labels. But for some other semi-supervised method, i.e., DividMix, although it performs better than many transition matrix based methods, this method is a combination of many existing technics. Therefore, we also integrate our proposed method into the DividMix method to further improve their performances, and final obtain the state-of-the-art results.

---

> > ### Comment · Reviewer_3zVp · 2022-08-06
> > **Response to rebuttal**
> >
> > I would like to thank the authors for their detailed rebuttal. While the authors' response answers many of my questions, a few important points of my review have remained unaddressed. In particular, despite the good results reported on some datasets, I believe the manuscript, in its current form, fails to meet the requirements for effective scientific communication.
> >
> > - Important experimental details are missing (from either the main text or the appendix -- in fact, the paper does not have an appendix at all) e.g. what are the exact noise models used for the experiments,
> >
> > - It is impossible to assess the experimental evaluation since the paper does not provide details on how the hyperparameters of the baselines were chosen, or how the hyperparameters of the proposed method were chosen.
> >
> > - Finally, the paper is made difficult to read by the many typos and mistakes, but also by the unusual structure (e.g. the introduction contains the proof of the “theoretical result” (as the paper notes on line 188).
> >
> > For these reasons I decide to maintain my score and suggest that the authors improve the manuscript before it can be ready for acceptance.

---

> > > ### Author Response · Authors · 2022-08-07
> > > **Response to reviewer's (3zVp) comments**
> > >
> > > Thank you very much for your hard work and quick response for our reply.  Here, we would like to answer the remaining questions to further address the reviewer's concerns.
> > >
> > > 1、We have updated the supplementary material of our paper, which include the code and appendix. In the appendix, we have clearly described the definition of the noise model, and the true transition matrix $T$  for three commonly used noisy types : symmetry flipping, asymmetry flipping, and pair flipping.  We highly encourage the reviewer to check our supplementary material to find these details, and thank you again for you hard work.
> > >
> > > 2、In the experiment part, we have provided the implementation details, including all the hyper-parameter settings of our proposed method.  In the  ablation study part, we have analyzed the important parameter $\lambda$ to explore its effect on model performance. Besides, we also provide the code in the supplementary material to help the readers to reproduce and evaluate our work.
> > >
> > > Specifically, the details of how the hyperparameters  are chosen can also be described here. '' For CIFAR-10 and CIFAR-100, the backbone we used is ResNet-34. We train the classification network $f(\textbf{x}_i;\textbf{w})$, the transition matrices $T$ and $T^{b}$ by SGD strategy, with batchsize of 128, momentum 0.9, weight decay $10^{-3}$, and learning rate $10^{-2}$. For CIFAR-10, the algorithm run 60 epochs and the learning rate is divided by 10 after the 30-$th$ epoch. For CIFAR-100, the algorithm run 80 epochs and the learning rate is divided by 10 after 30-$th$ and 60-$th$ epoch.
> > > For Clothing 1M and Food-101N, the backbone we used is ResNet-50 which is pre-trained on ImageNet. We train the classification network $f(\textbf{x}_i;\textbf{w})$, the transition matrices $T$ and $T^{b}$ also with SGD strategy, with batchsize of 32, momentum 0.9, weight decay $10^{-3}$, and learning rate $2 \times 10^{-3}$. The algorithm run 80 epochs and the learning rate is divided by 10 every 30 epochs. Before training, we warm up  on all noisy data with early stopping technique, where we have trained 10, 10, 1 and 1 epochs on the CIFAR-10, CIFAR-100, Clothing 1M and Food 101N datasets, respectively.  ''
> > >
> > > While for the parameters of the baseline methods,  our reported results are based on the public code provided by the authors, and each number in all the tables is the mean of five runs. We compared all the methods under the same experiment settings for fair comparison, including the baseline network architecture, noisy types, etc. All the hyper-parameter setting strategy can be found in the corresponding references or code, and we have further checked all the important references in our paper  to help the readers to assess the experimental evaluation.
> > >
> > > 3、Thank you very much for pointing out some typos and mistakes in our paper.  Last few days, we have carefully proofread the manuscript for several times,  and we would like to update the manuscript if needed in the future. We also found that these few typos  do not affect the readers to understand our paper clearly, since the other three reviewers give us positive comments  and one reviewer points out that our paper is well-written. For the  unusual structure in this paper (e.g. the introduction contains the proof of the “theoretical result”),  we aim to help the readers understanding our method in more depth and clearly, since the theoretical result is not complex to be proved.
> > >
> > > We sincerely hope that our responses can address all the remaining concerns. Above all, we really hope that our paper could be reconsidered and evaluated.

---

> > > > ### Comment · Reviewer_3zVp · 2022-08-08
> > > > **Response to rebuttal**
> > > >
> > > > Thank you for the prompt response. Regarding the second point, my question was about how the hyperparameter values were chosen, and not what the exact values are. I saw the paragraph in the main text listing the hyperparameter configuration, but this does not clarify how the authors arrived at these values.
> > > >
> > > > Minor remark: I think there is a typo in eq 1 in the appendix; the off-diagonal terms are probably \eps/(C-1)?

---

> > > > > ### Author Response · Authors · 2022-08-08
> > > > > **Response to reviewer's (3zVp) comments**
> > > > >
> > > > > Thank you very much for your quick responses. Your comments have greatly helped improve the quality of our paper. Here, we would like to answer the remaining questions.
> > > > >
> > > > > 1、In the appendix, we have corrected the typo in equation 1, and we have updated the supplementary material. Thank you very much for pointing out this typo.
> > > > >
> > > > > 2、For the questions of how the hyper-parameters were chosen,  we divide the hyper-parameters we have used in our paper into two categories, to answer this question.
> > > > >
> > > > > The first one is the most important trade-off hyper-parameter $\lambda$ in Eq.(5). We use the greedy search  strategy to choose its best value, where the search interval is [0.0, 1.0], and the search step is 0.1.  As shown in Figure 2(m) to (p), we evaluate different values of $\lambda$ on the CIFAR-10 and CIFAR-100 datasets, with noise rate 0.4 under the ''pair flipping'' noisy type. We have found that we can obtain the best performances on both datasets  when $\lambda$ is 0.3.  Therefore, we set $\lambda$ as 0.3 in all our experiments including the two synthetic and two real-world datasets.
> > > > >
> > > > > The second category is about the parameters used during model training, including the network architecture,  optimization method, batchsize, momentum, weight decay, learning rate, number of training epochs, etc..  All these parameters are set the same as our compared method VolMinNet [13], which provided the source code and the corresponding best parameters.  When we combine our method with the DivideMix [11] method, these optimization parameters are set  the same as DivideMix [11], which also provided the source code.
> > > > >
> > > > > We sincerely hope that our responses can address all the remaining concerns. Thank you again for your great help and many good questions and suggestions, which largely help improve the quality of our paper. We would like to clarify if you have further concerns. We really hope that our paper could be reconsidered by the reviewer.

---

> > > > > > ### Author Response · Authors · 2022-08-10
> > > > > > **Concerns addressed**
> > > > > >
> > > > > > Dear reviewer 3zVp,
> > > > > >
> > > > > > It seems we have addressed all your major concerns. Can you kindly reconsider the recommendation? Thanks very much.
> > > > > >
> > > > > > Best

---

> ### Author Response · Authors · 2022-08-02
> **Response to Reviewer's (Review3zVp) comments (Part 2)**
>
> 4、How does a simple baseline that uses early stopping regularization compare to the proposed method?
>
> We compared our method with another recently proposed early stopping method for label-noise learning on the real-world dataset Clothing1M, since the referenced two do not provide proper code for reproduction. We can clearly see that our method also outperforms such method with early stopping regularization. Furthermore, if we combine our method with these technics, I believe it could further boost the performance of our method.
>
> | Dataset  | Clothing1M |
> | Method  | CE | ForwardT[20] | VolMinNet[13] | DivideMix[11] |  ERL[15] | EarlyStop | Ours|   DivideMix+VolMinNet |  DivideMix+Ours |
> | :-----------: | ----: | ----: | :----: | ----: | ----: | :----: | ----: | ----: | :----: |
> | Accuracy | 68.94% | 69.84% | 69.82% |  74.67% | 72.87%| 74.64% |  70.73% | 74.83% | 75.12%$\pm$0.05$ |
>
> ERL[15]: Early-Learning Regularization Prevents Memorization of Noisy Labels, NIPS 2020.
>
> EarlyStop: Understanding and Improving Early Stopping for Learning with Noisy Labels, NIPS 2021.
>
> 5、I suggest adding to the experimental comparison with more natural label noise datasets(e.g., OpenImage, MS-COCO, MegaFace).
>
> MS-COCO is a large image recognition/classification, object detection, segmentation, and captioning dataset, it contains 330K images with more than 2M instances in the 80 object categories. This dataset is a multi-label datasets with 5 objects per image. OpenImages is a dataset of ~9M images annotated with image-level labels, object bounding boxes, object segmentation masks, visual relationships, and localized narratives. For each image, it also have multiple labels.
>
>  However, our method mainly focuses on the traditional single-label classification problem at present. MegaFace Challenge 2 (MF2) is a real world long-tailed noisy dataset: Training on 672K identities and 4.7M photos, and tested at million scale. The main challenge for this dataset is designed to cope with the long-tail distribution data with openset identities. In the future, we will extend our method on these datasets to cope with the multi-label classification problem with noisy label, and also the long-tailed data distribution problem.
>
> 6、What are the exact noise models? How many classes are assumed to suffer from label noise in the symmetric/asymetric/pair model?
>
> The exact noise models follow the traditional experiment settings, we need to manually corrupt these datasets by the noise transition matrix $T$, where $T_{ij}(\textbf{x})=P(\bar{Y}=j|Y=i,X=\textbf{x})$. It means that we have the noisy label $\hat{y}_j$ which is flipped from the clean label $y_i$.  In all the experimental setting,  we assume that all the classes in the dataset suffer from label noise in the Symmetry/Asymmetry/Pair fliping model.
>
> Specifically,  the Symmetry fliping: replacing the original label of each sample image with a probability, or with other class labels randomly drawn from the class set.  Asymmetry flipping: according to the similarity of the class, all samples of a class are generated by flipping all samples of a class to a specific class label similar to it with a probability (such as bird <-> airplane, cat <-> dog, deer <-> horse, etc.). Pair flipping: a simulation of fine-grained classification with noisy labels, where labelers may make mistakes only within very similar classes.
>
> 7、The method seems to not perform particularly well on datasets with natural noise. Why are only some and not all baselines presented in tables 4 and 5? What are the confidence intervals for the values in tables 4 and 5?
>
> Although we do not list all previous label-noise learning methods here, in reality we have compared 14 recent representative works on the Clothing1M dataset in Table 4, which almost contain most of the representative statistically consistent algorithms and some semi-supervised methods. While on the Food-101N dataset, we almost compared our method with all the previous method including the statistically consistent and inconsistent algorithms, and we get state-of-the-art performance.
>
> For the confidence intervals of the values in table 4 and 5, we have done detailed survey about the previously reported experiment results on these two datasets, even there is no algorithm ever report the confident interval on these real-world datasets. Because the noise distribution, noise ratio are all fixed, all the algorithms just report the final classification accuracy. But for the synthetic datasets, we always report the confidence interval since the synthetic noise is randomly generated in each running.
>
> However, in order to meet the requirement of the reviewer, we also report the mean classification accuracy and the standard deviation computed over five runs on these two real-world datasets as follows.
>
> | Method  | Cloting1M | Food-101N|
> | :-----------: | ----: | :----: |
> | DivideMix | 74.58 | 84.37 |
> | DivideMix+ours | 75.12$\pm$0.05 | 86.11$\pm$0.03 |

---

> ### Author Response · Authors · 2022-08-02
> **Response to Reviewer's (Review3zVp) comments (Part 3)**
>
> 8、It is unclear from the results reported in the tables (in particular Tables 2 and 3) when the proposed method breaks and what exactly are the assumptions that it requires to perform well.
>
> The exactly requirement of the assumption is to utilize as many as confidient examples during model training. In oder to reveal  what will happen if the assumption breaks in the proposed method, we design the following experiments to make detailed comparison between our proposed method and the baseline method VolMinNet.  In the experiments, we first train a classifier with clean labels on the Cifar-10 dataset. Then we utilize this model to remove 50% confident examples of every class in the synthetic noisy dataset. After that, we train our proposed method and the VolMinNet with the same synthetic noisy subset. Finally, we test these two model on the same testing dataset. The experiment results are listed below, we can clearly see that our algorithm more robust compared with the baseline method VolMinNet.
>
> | Dataset  | Cifar-10( Synthetic Noisy Subset) |
> | Method  | Sys-20 | Sys-40 | Sys-60 |
> | :-----------: | ----: | ----: | ----: |
> | VolMinNet [13] | 85.34$\pm$0.08 | 80.33$\pm$0.12 | 61.48$\pm$0.58 |
> | ours | 86.06$\pm$0.06 | 81.40$\pm$0.23 | 71.53$\pm$0.29 |
>
> 9、Instead of selecting the 3 fixed noise rates reported in the paper, I suggest a plot in which the noise rate is varied on the Ox axis, while the Oy axis indicates the test accuracy.
>
> In reality，selecting 3 fixed noise rates is the common experiment settings  in  the label-noise learning research field,  most of the research works follow this experiment setting as reported in our paper. In order to give more detailed analysis of the experiment results as the reviewer suggested, we have also done more experiment with different noise ratios. Given the time issue, we just did experiments on Cifar-10 dataset with 6 noise ratios, and the experiment results demonstrate the same conclusion as reportaed in the paper.
>
> | Dataset  | Cifar-10 |
> | Method  | Sys-10 | Sys-20 | Sys-30 | Sys-40 | Sys-50 | Sys-60 |
> | :-----------: | ----: | ----: | ----: | ----: | ----: | ----: |
> | ours | 91.54$\pm$0.13 | 90.44$\pm$0.19 | 88.98$\pm$0.15 | 87.30$\pm$0.25 | 84.18$\pm$0.12 | 81.01$\pm$0.25 |

---

> ### Author Response · Authors · 2022-08-06
> **Response to Reviewer  3zVp**
>
> Dear reviewer  3zVp,
>
> Thank you very much for reviewing our paper and giving us some good questions.
>
> We have tried our best to answer all the questions according to the comments. We sincerely hope that our responses can address all your concerns. Is there anything that needs us to further clarify for the given concerns?
>
> Thank you again for your hard work.

---

### Author Response · Authors · 2022-08-05
**Comments to the AC and all reviewers**

Dear AC and all the reviewers,

Thanks for handling our manuscript.

We  have tried our best to answer all questions of the reviewers about our paper. We wander  if our responses address all the concerns ?

Thanks all !

---

### Meta-Review · Area_Chair_bRrd · 2022-08-23

**Recommendation:** Accept
**Confidence:** Less certain

**Metareview:**

This work addresses the problem of estimating the transition matrix by using forward-backward cycle-consistency, with class-dependent noisy labels. There is merit in this work, as the proposed method might encourage the estimated transition matrix to converge to its optimal solution, without explicitly estimating the noisy class posterior probability. Therefore, it could help to build better statistically consistent classifiers.  It is shown theoretically that the proposed method is superior compared to compared methods, and the effectiveness of the method is demonstrated on several different datasets. There was a lively discussion going on between the reviewers and the authors. Although there remain some open questions about how the hyperparameter values are chosen, I think this paper should be accepted.

**Award:**

No

---

### Decision · Program_Chairs · 2022-09-14

Accept